# Introducing and Familiarising Older Adults Living with Dementia and Their Caregivers to Virtual Reality

**DOI:** 10.3390/ijerph192316343

**Published:** 2022-12-06

**Authors:** Aisling Flynn, Marguerite Barry, Wei Qi Koh, Gearóid Reilly, Attracta Brennan, Sam Redfern, Dympna Casey

**Affiliations:** 1School of Nursing and Midwifery, University of Galway, H91 TK33 Galway, Ireland; weiqi.koh@universityofgalway.ie (W.Q.K.); dympna.casey@universityofgalway.ie (D.C.); 2Information and Communication Studies, ADAPT Centre, University College Dublin, D04 V1W8 Dublin, Ireland; marguerite.barry@ucd.ie; 3School of Computer Science, University of Galway, H91 TK33 Galway, Ireland; g.reilly6@universityofgalway.ie (G.R.); attracta.brennan@universityofgalway.ie (A.B.); sam.redfern@universityofgalway.ie (S.R.)

**Keywords:** dementia, older adult, virtual reality, VR, technology probe, participatory methods, human–computer interaction, gerontechnology, AgeTech, digital technology

## Abstract

Virtual Reality (VR) is increasingly being applied in dementia care across a range of applications and domains including health and wellbeing. Despite the commercial availability of VR, informants of design are not always aware of its functionality and capabilities, to meaningfully contribute to VR design. In designing VR applications for people living with dementia, it is recommended that older adults living with dementia and their support persons be involved in the design process using participatory approaches, thereby giving them a voice on the design of technology from the outset. A VR technology probe is a useful means of familiarising older adults living with dementia and their informal caregivers with the knowledge and understanding of interactive VR to employ technology that supports them to maintain their social health. This paper charts the implementation and evaluation of a VR technology probe, VR FOUNDations. To explore their experiences, nine older adults living with dementia and their nine informal caregivers trialled VR FOUNDations and completed semi-structured interviews after its use. Overall, older adults living with dementia and their informal caregivers perceived VR FOUNDations to achieve its aim of increasing understanding and inspiring future design decisions. The findings also identified promising positive experiences using a VR technology probe which may be indicative of its applicability to social health and wellbeing domains. This paper advocates for the structured design and implementation of VR technology probes as a prerequisite to the participatory design of VR applications for the health and wellbeing of people living with dementia. The use of such technology probes may afford older adults living with dementia and their informal caregivers the best opportunity to contribute to design decisions and participate in technology design to support their health and wellbeing.

## 1. Introduction

Dementia is a degenerative condition which results in a decline in one’s cognitive function and leads to changes in one’s daily functioning and social interactions [1]. Although the presentation and lived experience of dementia are vastly individualised and dynamic, dementia more commonly impacts cognitive and physical functioning and may include reduced memory recall, attention, verbal abilities, and visual representation [2,3]. Given that there is currently no cure for dementia [4], digital technology is increasingly being used as a means of enhancing the quality of life for people living with dementia throughout the disease trajectory [5,6,7]. However, the adoption of mainstream digital technology such as Virtual Reality (VR) has generally been low for people living with dementia due to a lack of awareness, accessibility, and support [8,9].

VR is growing in prevalence in the Human-Computer Interaction (HCI) community [10] and has a myriad of applications in dementia care contexts [10,11,12,13] including residential aged care facilities (RACF), acute inpatient settings, community day centres and home environments [14]. Traditional non-pharmacological interventions such as Reminiscence Therapy and Cognitive Stimulation Therapy are also being successfully incorporated into VR applications [15,16,17,18,19]. VR has shown promise in supporting the emotional, social, health and functional wellbeing of people living with dementia. However, most research to date has been exploratory [13,14,20]. A recent focus has been the use of Social VR to promote social health outcomes [13,14,20]. COVID-19 accelerated this shift in focus towards social technologies as physical distancing measures increased feelings of social isolation and loneliness for older adults with and without dementia [21,22]. In such circumstances, VR can provide a means of staying socially connected and can facilitate engagement in meaningful activities for people living with dementia [14]. Due to its relatively novel commercial availability and previously high cost to acquire, older adults typically do not have direct experience of VR [23,24]. Researchers advocate for the sensitive and graded introduction of VR for this population [23,24,25,26].

When designing technology to support the health needs of this population, accessibility and suitability to the individual’s needs must be prioritised [6,27,28,29,30,31]. The drive toward participatory approaches warrants the inclusion of people living with dementia and their informal caregivers as ‘experts-by-experience’ to plan for data collection and identify design priorities for VR applications [32,33,34,35]. Despite the benefits of involving people living with dementia and other stakeholders in the technology design process, Muñoz et al. [10] acknowledge low adoption rates for these studies and a lack of consensus on the most appropriate methods to support their involvement. Appel et al. [13] also acknowledge a paucity of research on how VR can promote the wellbeing of people living with dementia including best practices for administration, evaluation methodologies and best hardware and software to be deployed.

The purpose of this study is to evaluate the use of a technology probe, VR FOUNDations (Virtual Reality Familiarisation envirOnment for older adUlts with aND without dementia). The aim of VR FOUNDations is to familiarise people living with dementia with a generic virtual environment (VE) and better position them to contribute to the design of future VR applications for health and wellbeing. In the case of this research project, the goal was to inspire the design of a bespoke VR application to enrich the social connectedness of older adults living with dementia and their caregivers. According to Gaver et al. [36], technology probes are used as a means to “address the challenge of designing new technologies for unfamiliar groups” cited in [37] (p. 920). In this study, a VR technology probe is necessary to enable older adults living with dementia and their caregivers to familiarise themselves with the functionalities of VR, so they are better positioned to suggest future design ideas. VR FOUNDations consists of a passive and an active VE (see Section 2.2 for details). The purpose of using both VEs is not to specifically explore the usability or feasibility of VR FOUNDations itself, but to understand how people living with dementia and their caregivers use VR and can inspire future design innovation [38,39,40,41,42,43]. The VEs are generic in design and serve as gateways to increasing understanding of VR, which in turn provides people living with dementia and their caregivers with a more informed voice regarding the design of future bespoke VR applications to support their health and wellbeing [38].

The motivation for the development of the VR technology probe came early in the conceptualisation of a larger VR and social health project, with an overall aim to design a VR social connecting space for older adults living with dementia. It became apparent that for people living with dementia and caregivers to make informed design decisions regarding VR for social health, they must be familiar with its general functionality and capabilities. The question then was how this relatively novel technology can be used as a prerequisite to participatory design methods to support and inspire future design ideas. In the case of this larger project, this prerequisite stage of inspiring innovation must be completed in technology and health domains to optimise the upcoming participatory processes. Examples of the use of such technology probes in dementia and health studies include the design of tablet technology for older adults [44], the creation of tangible artefacts to trigger reminiscence [40], VR-based stroke-rehabilitation [45] and the exploration of social VR experiences [41]. Despite the positive application of technology probes in these studies, there is a paucity of research on their use in the VR and dementia care landscape, thus making it difficult for researchers to effectively design and implement them.

This paper explores the experiential aspects of using VR FOUNDations, a VR technology probe, as a tool to increase understanding and elicit discussion on the design of a future bespoke VR application for social health. The research was driven by the following research questions:
How do older adults living with dementia and their caregivers experience a VR technology probe?○Can a VR technology probe increase understanding of the basic capabilities of VR?○Can a VR technology probe inspire the future design of a bespoke VR application for social health?How can VR technology probes be designed and implemented in dementia care research?


## 2. Materials and Methods

### 2.1. Study Design

This fieldwork is the first phase of a larger Participatory Action Research (PAR) study which aims to explore the design of a VR social connecting space for people living with dementia over 60 years of age. This research is focused on how to empower and emancipate people living with dementia and their caregivers to participate in research activities and subsequent data collection phases [25,41]. PAR has been applied in several dementia care studies to actively involve people living with dementia and their caregivers in the research process [46,47,48,49]. PAR as a methodology provides a means of privileging the voices of older adults living with dementia and their caregivers. This enables them to guide the research direction and is advocated by Baker and colleagues [41] and the wider HCI research community [50]. Studies show that the use of technology probes complement PAR by empowering people living with dementia and their caregivers to meaningfully have their voices heard [51].

The methods used in the design of the VR FOUNDations report on one action research cycle of the larger study. People living with dementia (>60 years of age) and their nominated caregivers involved in this cycle formed part of a PAR group that agreed to be involved for the duration of the larger project across multiple PAR cycles. Data were gathered through observational field notes, followed by separate semi-structured interviews with people living with dementia and their caregivers, in order to explore their experiences of using the VR technology probe and to understand whether they perceived it to increase their understanding and inspire future VR design for social health. The PAR cycle in this study involved developing the technology probe (planning), trialling the technology probe, holding interviews (acting) and compiling design considerations that may inform the design of a VR application for social health and wellbeing (reflecting). The findings from this research will then inform subsequent action research cycles aimed at developing a VR application to enhance the social connectedness of older adults living with dementia. Ethical approval was sought and granted by the University of Galway Research Ethics Committee (reference number 2021.03.007).

### 2.2. Design and Implementation of VR FOUNDations

The technology probe consisted of a commercially available passive VE using Oculus library environments and a bespoke active environment. For this study, a passive VE was considered one “where the user is only watching” while the active VE is one where the user is “actively involved” [52] (p. 86).

While there are commercial active VR applications that can be utilised, they are not specifically designed as technology probes for older adults or people living with dementia. As put forward by Dixon and Lazar [28], technology for people living with dementia should support activities that are meaningful to them in a way that respects and reflects their needs, abilities, and perceptions of personhood. As people living with dementia have wide and dynamic preferences and abilities, it is crucial that their voices are central to the design process of technologies intended for their use. Furthermore, the lead researcher, who is a qualified Occupational Therapist (OT), using her professional judgement, considered the visual and auditory content of commercial active alternatives to be overly stimulating and complex. In addition, the interactions within VR were not adaptable, which contradicts the open-ended and exploratory nature of technology probes proposed in the literature. Jong et al. [48] noted that technology probes may help set expectations for future bespoke environments from the outset; something that repurposing commercial VR applications (such as Oculus ‘First Steps’ or ‘First Contact’) may not afford.

Despite no suitable commercially available active application, the passive Oculus ‘lobby’ VEs were commercially available and considered appropriate for people living with dementia and older adults. These lobby environments cannot be manipulated and are simply used to passively observe the VE. The lobbies are a set of passive environments that are pre-loaded and set as the Oculus home screen. They consist of a range of environments such as a space station, ski lodge, or yoga studio. The research team coupled the available passive library of lobbies and aimed to address the lack of a suitable active VE by designing one to accompany them. Both the active and passive VEs contributed to the technology probe and enabled a graded experience from the passive VE to the active which is advocated for people living with dementia and their caregivers [14].

The research team, which comprised an Occupational Therapist (OT), a Health Psychologist and a Games Developer iteratively and reflexively developed the active VE. Two people living with dementia who formed a Public and Patient Involvement (PPI) advisory group, participated in the initial design and pilot of the technology probe, VR FOUNDations. Details of the design process and role of PPI members will be reported in a separate publication.

The active environment in VR FOUNDations consisted of one large room with several stations, to which people living with dementia and their caregivers could navigate and experience different modes of interactions (see Figure 1,Figure 2 and Figure 3). Table 1 provides an overview of the various interactions and their implementation in the active VE. These interactions were chosen to reflect three universal tasks of navigation, selection and manipulation [53]. Such tasks were considered a means of exposing people living with dementia to the basic interactive features of VR for the first time, including grasping, picking up and repositioning items, in addition to navigating in the VE. While such interactions are used in commercially available VR applications such as ‘First Steps’, this study aimed to design these tasks specifically for the needs of older adults and people living with dementia. The design of the interactive tasks and content in the VE were considered better suited to the needs of older adults living with dementia as they provided clear visual instructions over each station, utilised contrasting colours throughout [54], had clear navigation cues (Figure 1), used familiar items and visually represented where one was pointing in the VE. Such design choices were not collectively provided by commercial alternatives. Appendix A provides a video presentation of VR FOUNDations.

VR FOUNDations adapted the Oculus Quest 2 controllers to allow for interactions such as pointing and clicking in the VE through one main ‘trigger’ button. To further facilitate interaction in the VE, a virtual representation of one’s hands were included (Figure 4). This was decided through consultation with the PPI advisory group members and is supported by the work of Karaosmanoglu et al. [55] and Abeele et al. [54], who found that controllers with multiple buttons were perceived as confusing for some older adults and people living with dementia.

### 2.3. Recruitment and Sample

The PAR group consisted of community-dwelling people living with dementia (*n* = 9) and their nominated caregiver (*n* = 9). The PAR group was recruited as part of a long-term PAR project which aimed to explore the design of a VR social connecting space. Group members consented to participate in the study over a 14–16-month period. Given the individual and dynamic manifestations of dementia and its impact on one’s ability to express preferences in some instances, the inclusion of their caregivers was considered appropriate [32]. Having caregivers involved in the study provided a sense of reassurance for people living with dementia and ensured that a holistic view of the lived experience of dementia could be achieved [55].

Due to the closure of day centres and in-person dementia services owing to the COVID-19 pandemic, recruitment was completed virtually through online memory cafés and “TeamUp for Dementia Research” (a national research database facilitated by the Alzheimer Society of Ireland) [56]. Formal written and verbal consent was obtained from people living with dementia and their caregivers at the start of the study and ongoing process consent was used throughout the data collection activities. The lead researcher used her clinical experience as an OT in dementia care and time spent with people living with dementia and their caregivers to establish capacity and gain informed consent. All people living with dementia were over 60 years of age (except for one person living with dementia who turned 60 years during the project) and had either a diagnosis of dementia via their GP, i.e., prescribed dementia-specific medication or a formal clinical diagnosis to be eligible for inclusion. The researcher did not discriminate between the stage of dementia but took a functional and strengths-based approach, whereby people living with dementia were eligible for inclusion if they had the physical and communication abilities to participate in data collection activities. The small sample size was considered appropriate given the constraints associated with recruitment during COVID-19 whilst also acknowledging the complexities associated with recruitment difficulties in dementia research [55].

### 2.4. Demographic Information

A questionnaire was completed during the home visit to gather demographic information from people living with dementia and their caregivers before trialling VR FOUNDations. The age range of people living with dementia extended from 59 to over 80 years of age. Most people living with dementia were male (*n* = 7), with only two females participating in the study. Table 2 and Table 3 present further demographic information, with a more comprehensive table in Appendix A. It is noteworthy that all people living with dementia and their caregivers were relatives (Table 2 and Table 3). Each person living with dementia had mild to moderate dementia based on a functional-based stage classification approach from the National Institute on Aging [57]. This approach to classification followed the work of Wood et al. [58] and the wider HCI Community [59], who acknowledged that when investigating opportunities for increasing technology experiences with people living with dementia, functional capacity is reflective of one’s ability to complete daily functional tasks.

When people living with dementia and their caregivers rated their familiarity with everyday technology and familiarity with VR (Table 4), 14 people living with dementia and their caregivers stated that they were familiar with technology, three people living with dementia reported some experience with technology, whilst one person living with dementia reported no experience with technology. In terms of VR experience, five people living with dementia and their caregivers stated they had seen VR but not used it personally, seven people living with dementia and their caregiver had tried passive VR (involving less interactive headsets) several years ago, whilst six people living with dementia and their caregivers stated that they had no prior knowledge of VR.

### 2.5. Methods of Data Collection

One home visit was collectively completed with each person living with dementia and their caregiver and lasted approximately two hours. The home visit included people living with dementia and their caregivers using VR separately, followed by individual interviews which lasted an average of 14.5 minutes. Semi-structured interviews were used to explore people living with dementia and caregivers’ experience of VR FOUNDations. In addition, detailed field notes (guided by an observational template (Appendix A)) were collected during use. This was used to gather data relating to people living with dementia’s use of VR in particular noting verbal and non-verbal responses, length of use, body positioning and tolerability (Appendix A). Directly after using VR FOUNDations, separate semi-structured interviews were conducted with people living with dementia and their caregivers. The interview schedule focused on people living with dementia and their caregivers’ experience of using VR FOUNDations (Appendix A). Although the experiences of caregivers were considered important, the experiential elements of VR use from the perspective of people living with dementia were the primary focus of this study.

### 2.6. Procedure

People living with dementia and their nominated caregivers participated in the trial of VR FOUNDations individually in their home environments, facilitated by the lead researcher. Initial informal orientation consisted of explaining the session aims and instructions on how to wear the headset and operate the controllers. This served as a means of establishing rapport before using VR. To minimise risk and ensure the safe use of the VR equipment, an eligibility checklist (Appendix A) and a distress protocol (Appendix A) were followed. People living with dementia and their caregivers were informed of the procedure and the possible negative implications of VR use (e.g., motion sickness, overreaching). People living with dementia and their caregivers were encouraged to ask questions relating to the set-up and use of VR FOUNDations before its use. They were made aware that the headset could be taken off at any point during use without consequence.

Although all people living with dementia and their caregivers had the opportunity to separately use VR FOUNDations, all people living with dementia used the technology probe prior to their caregivers, aside from one male living with dementia (PwD7). Caregivers were amenable to have the person living with dementia use VR FOUNDations first as some were apprehensive to use the technology for the first time. People living with dementia and their caregivers followed the implementation procedure as outlined in Section 2.2, starting with the passive environment, and transitioning into the active. They also had the option to use VR FOUNDations either seated or standing. The researcher and caregivers were present when people living with dementia used the headset. To ensure adequate assistance, the researcher could view what people living with dementia and their caregivers were seeing through an iPad. Assistance consisted of verbal guidance, re-orientation to controller buttons or physical body repositioning during use. The researcher used her clinical experience as an OT in dementia care to guide facilitation, using clear communication strategies and being observant of verbal and nonverbal responses throughout VR use in their homes.

After 20–25 minutes, people living with dementia or their caregivers were asked to finish using the technology and return to the physical space. This time limit is consistent with the existing VR literature in the field of gerontology and dementia care [14,60].

### 2.7. Data Analysis

All data were transcribed verbatim, and each transcript was sent to the respective person living with dementia and their caregivers for member checking. Any necessary changes were made as requested (these consisted of additions to clarify the meaning/provide context to statements made or verify assumptions made by the researcher, etc.). Observational field notes and interview transcripts were analysed using reflexive thematic analysis (TA) as per Braun and Clarke [61,62]. This involves six stages: (1) familiarising oneself with the data, (2) generating codes, (3) constructing themes, (4) reviewing potential themes, (5) defining and naming themes, and (6) producing the report [61,62]. An inductive approach to data analysis was taken whereby codes and themes were data-driven as opposed to ‘fitting’ them into a predefined framework [61,62].

The researchers were aware of their positionality throughout the data collection process and took a critical realist stance to TA; “to provide a coherent and compelling interpretation of the data, grounded in, or anchored by, the participant’s accounts, that speaks to situated realities” [63] (p. 171). TA provided a means of placing the lived experience of people living with dementia and their caregivers’ at the forefront while also exploring the resources and understandings that underpin their accounts [63].

NVivo 20 was used to manage the stages of data analysis. Data from people living with dementia and their caregivers were coded separately and merged at Stage 4 of data analysis. The initial discussion of preliminary themes/defining involved two researchers (AF and WQK). This was followed by an informal meeting with people living with dementia and their caregivers to discuss these themes and to reflect on the data collection process. People living with dementia and caregivers agreed with the derived themes and noted that they were representative of their experience of VR FOUNDations. This approach also ensured the rigour and authenticity of the data and set the agenda for subsequent PAR cycles.

### 2.8. Reflexive Statement

Interviews and informal meetings were completed by AF, a doctoral student and OT. AF has clinical experience working in dementia care contexts and Memory Technology Resource Rooms. AF had no pre-existing relationship with the PAR group members prior to this project. WQK (OT) assisted with data analysis. She has clinical experience working with people living with dementia in different health and social care settings and research experience exploring the use of technology to support the social health of people living with dementia.

Throughout the data analysis process, AF and WQK discussed the transcripts, the coding process, and the derivation of themes. They also discussed how their positionality as OTs influenced the findings. AF and WQK were mindful of the functional capacities of people living with dementia and their caregivers and how this impacted their experience of using VR. AF and WQK brought their clinical experience of working in dementia care when analysing the data. They explored people with dementia’s experiences beyond their diagnoses, to explore the interplay between their physical, mental, and social functioning and how these influenced their experience of VR.

The findings reflect the person-centred and strength-based approaches advocated in OT practice. To minimise researcher bias, a summary of the findings was presented to people living with dementia and their caregivers [64]. AF also kept a reflexive journal throughout the research process to identify how her positionality may influence the data analysis and reporting processes. 

## 3. Findings

Four themes were generated from the data analysis, including the impact of multisensory VR, adapting and accommodating autonomy, providing real assistance for a virtual experience and dissipating apprehensions through exposure and understanding (see Figure 5). The findings suggest that VR FOUNDations was positively experienced by people living with dementia and their caregivers and can be used as a prerequisite for the design of health and wellbeing VR applications for people living with dementia.

### 3.1. The Impact of Multisensory VR: From Alertness to Relaxation

This theme describes how the use of VR provided a means of multisensory stimulation for people living with dementia and their caregivers through visual, audio and haptic stimuli. It also references how being immersed and present within VR FOUNDations led to increased alertness and engagement. The multisensory features such as the three-dimensional display, 360-degree view, realism and aesthetic presentation of the VEs were referenced as contributing to engagement and alertness. Verbalisations captured the sense of awe and excitement attributed to the visual stimuli in the VE: “Never seen anything like it before” [PwD7] and “Marvellous..so realistic” [PwD9]. The multisensory and generic content, albeit not designed for personalisation, scaffolded reminiscence, whereby people living with dementia related the content to previous holiday destinations and occupations and shared these memories with the researcher and their caregivers. The haptics, dexterity, interactivity and responsiveness of the virtual hands led to a sense of control and subsequent embodiment in the VE.

The sense of presence afforded by the multisensory VE also contributed to increased alertness, engagement and interaction during VR use for all people living with dementia and caregivers. One person living with dementia appeared lethargic preceding VR use and was observed to be more attentive once the headset was put on. Similarly, those who were initially apathetic were observed to be smiling, laughing and socially engaged during VR use. This increased alertness and engagement were reported as a welcome surprise for the caregivers. A sense of “getting lost in it [the VE]” [PwD4; CG5] and a sense of flow were alluded to by two caregivers: “he was going with the flow” [CG2], illustrating enjoyment and engagement in the VE. The positive reports of the caregiver are comparable to observational data from researcher field notes and direct accounts from people living with dementia after using VR FOUNDations.

VR was reported as having a “calming” [PwD7] and “relaxing” [PwD2] effect for some people living with dementia. When using VR FOUNDations, people living with dementia and caregivers were observed to be at ease, not seeking mastery of tasks but simply enjoying the new experience. While the design of VR FOUNDations required skills to complete tasks, this was not perceived as a “test” [PwD7; CG8]. People living with dementia described it as a safe space to foster fun and free exploration. This sense of fun was expressed by those who completed tasks with ease in addition to those who required more assistance.

In contrast to the positive affordances of multisensory VR, some people living with dementia and caregivers experienced negative reactions and verbalisations. The multisensory VE was perceived as overstimulating, lacking consistent clarity or demanding too much concentration. One caregiver indicated that the inconsistent distance between virtual hands and objects led to a sense of disconnection from their own hands. Two people living with dementia also had a neutral experience and stated that “it was okay” [PwD5] and needing to get “used to” [PwD5] VR over time.

### 3.2. Adapting and Accommodating Autonomy

This theme describes the importance of adapting the use of VR FOUNDations to the individual needs of people living with dementia and the value of experiencing the spectrum of interactivity. By having both passive and active VEs, people living with dementia and their caregivers were given the autonomy to decide the level of interactivity they preferred.

Some people living with dementia expressed a preference to passively interact in the lobbies by naming items, swivelling on their chair and simply taking in the 360-degree view. Those who expressed such preferences demonstrated decreased physical functioning, with one female living with dementia ambulating with a walking aid. People living with dementia who easily managed the controllers expressed a preference for greater interactivity and free exploration to satisfy their curious nature and increase their agency. Such participants had no physical difficulties and were observed to have good standing or sitting balance. This acknowledged the individual preferences of people living with dementia when using VR FOUNDations, whereby they were given a choice over the level of interactivity and gauged it to their own needs.

No negative side effects were perceived by people living with dementia or their caregivers during or immediately after VR use. Some software and hardware issues were identified, specifically, the clarity of the graphics in the active VE was cited as “jittery” [CG5] by some users, while others complained of low sound volume and their glasses fogging up when used with the headset.

Mostly, the hardware and software appeared to be congruent with the functional abilities of people living with dementia and their caregivers. The headset was universally seen as positive and suited to the needs of people living with dementia. The ‘Oculus Comfort Head Strap’ and glasses spacer also enhanced the comfort for those wearing glasses. Those with arthritis, tremors, an ataxic gait, reduced standing tolerance or balance difficulties demonstrated an ability to use the controllers and the headset. The option to use VR FOUNDations in either a seated or standing position accommodated those with reduced standing tolerance and balance. The organisation of tasks in the active VE was described as intuitive and natural. This was attributed to the use of one main ‘trigger’ button on the controllers and the presence of ‘rays’ in the VE to show where people living with dementia and their caregiver were pointing. The majority of people living with dementia reported that they enjoyed the tasks and could complete them with varying levels of assistance. People living with dementia and their caregivers noted, and it was also observed in field notes, that the use of controllers and task completion in the VE improved with exposure. This suggests that some difficulties interacting in the VE may be dissipated with familiarisation and may not be attributed to one’s functional capabilities or diagnosis of dementia.

### 3.3. Providing Real Assistance for a Virtual Experience

This theme describes the important role of facilitation and gauging the level of assistance required for people living with dementia. It was evident that the dynamic needs and abilities of people living with dementia required individualised facilitation and assistance. It was important that such assistance did not undermine the capabilities of people living with dementia or diminish their opportunity for free exploration in the VE. Assistance and facilitation manifested in the form of simple verbal instructions, visual written instruction cues in the active VE, and physical assistance. The initial use of the passive environment and then the transition to the interactive environment was reported as helpful by people living with dementia and their caregivers. This graded and paced approach was in keeping with their level of familiarity with VR and was seen to be useful in this case. All people living with dementia had the cognitive capacity to attend to and understand verbal instructions from the facilitator and respond as necessary in the VE, which suggests that they were able to distinguish between the physical and virtual space. One caregiver reported that it was like being “between two realms” [CG1].

Verbal instructions were provided in a stepwise manner and people living with dementia were given time to respond before the next set of instructions “Your timing was enough for someone to listen, follow, do it. It was very good” [PwD9]; “Well, I mean, it’s just you, you explained everything […] your explanations were just wonderful […] You did not have to repeat anything. It was very straightforward” [CG7]. By using the casting function afforded by the Oculus Quest 2, the interactions of people living with dementia and their caregivers within VR were displayed in real time on an iPad. This enabled the researcher to facilitate and assist accordingly. Physical assistance was particularly pertinent when people living with dementia did not use a swivel chair and/or needed to be repositioned within the ‘guardian zone’ (an Oculus safety feature whereby you create a safe interaction zone to avoid hitting into/tripping over items in the physical space), or when people living with dementia’s fingers needed to be reorientated to the trigger button, to point or hover over items in VR FOUNDations. Although minimal prompting was required for the passive lobby VE, some people living with dementia required additional facilitation within the active environment due to its increased complexity.

Assistance was seen as mutually beneficial for the lead researcher and people living with dementia as they had the opportunity to explore the experiential elements of using the VR technology probe while people living with dementia appreciated the assistance. Caregivers also reported that their presence provided reassurance for people living with dementia while using VR FOUNDations. Having the facilitator physically present when using VR FOUNDations was also considered essential by people living with dementia and their caregivers. Verbal instruction by the researcher was considered more effective than visual and written instructions in the VE. This considered approach to assistance ensured that the abilities of people living with dementia were respected, allaying perceived concerns about using VR. One male living with dementia reported finding the instructions useful “anything that helps make life easier” [PwD3]. People living with dementia and their caregivers placed a level of trust in the facilitator as they knew about VR and its capabilities, “It’s just because you [researcher] have a good knowledge about it and what sort of things people do right or wrong” [PwD6], “You were there to nudge us in the right direction if we were getting a bit lost” [PwD7].

Three people living with dementia verbalised that their caregiver’s presence was useful while they were using VR in real time. This was also perceived as useful in the researcher field notes and caregiver accounts. Other people living with dementia and caregivers agreed that they did not require their caregiver’s presence when using VR FOUNDations.

When discussing using VR again in the future, people living with dementia highlighted that their caregivers would have to assist with the set-up and expressed a preference for an instruction manual or videos to accommodate future set-up, such as how to wear and remove the headset, controller orientation and how to launch the application. The importance of dementia-friendly and accessible documents was also highlighted. Troubleshooting was considered important for the caregiver, i.e., having somewhere they could go to look up common issues and how to resolve them. Having a VE that is designed to suit the preferences of not only people living with dementia but also their caregivers is essential to ensure that the latter’s role as co-facilitators of the technology is acknowledged: “I would not be very techie. However, for the knowledge that I do have about downloading an app, I think I would be able to manage that […] and then there’ll be instructions with it so, I think I would be okay to manage it.” [CG4], “ I just need [..] normal instructions” [CG9]. Therefore, the VR design features and activities need to be of mutual interest to both people living with dementia and their caregivers, and graded to suit all of their needs. Interestingly, the main contributors to this theme were caregivers.

### 3.4. Dissipating Apprehensions through Exposure and Understanding

This theme describes the initial perceived challenges relating to VR use and the impact of exposure to VR FOUNDations in changing such perceptions. The novelty of VR led to perceived challenges, whereby people living with dementia were concerned about their ability to engage with it. Caregivers and people living with dementia reported a perceived level of anxiety and uncertainty as to how people living with dementia would tolerate VR, “am I going to be able to manage it” [PwD1] or “what would I be exposed to” [PwD7]. Before use, people living with dementia and caregivers associated VR with arcade games or Zoom and some users were concerned about using the technology as they were not considered gamers. Those who embraced VR FOUNDations reported fewer apprehensions and stated they were excited and curious to try. Interestingly, people living with dementia between the age of 70–79 years reported the most preconceptions and anticipation associated with the use, while their caregivers reported anticipation and concerns across all age ranges. First-hand experience of using a technology probe enhanced people living with dementia and their caregiver’s understanding of VR and addressed their preconceptions. Some of people living with dementia and their caregivers reported gaining an “idea of what it was like” [PwD5] and knowledge of its interactive capabilities which were afforded by the spectrum of interactivity.

Understanding VR sparked interest in its future use for people living with dementia. Future interest was attributed to the innovative nature of VR and/or their self-reported curious personalities. In contrast, others stated that they would like to use it again provided certain design elements were amended e.g more audio/background noise and adding smell or touch. One male stated that he would not use it again as “you might see things you do not want to see” [PwD4]. He further expanded on this statement to say that it is like the movies, you may not have an interest in what you are viewing. Facilitating understanding using VR FOUNDations enabled families to inform future design as they are provided with opportunities to make design recommendations and outline activity preferences for further VR experiences. Travel, flight simulation, sport, music, games, and relaxation were all listed as possible ideas for future use.

## 4. Discussion

The four themes not only reveal some interesting insights into the use of technology probes, specifically VR technology probes, with people living with dementia and their caregivers but for the wider HCI community. Each theme will be discussed and positioned within existing literature in the HCI, gerontology and dementia care landscape.

Although the purpose of VR FOUNDations was to expose people living with dementia and their caregivers to VR and gain an understanding of the basic interactive capabilities of VR to inform future use, people living with dementia self-reported positive experiential elements to use consistent with wider dementia and gerontology literature [13,14,54,65,66]. This included enjoyment and fun, alertness, increased engagement and verbalisations and relaxation. Such responses highlight the promise of generic VR experiences to increase the social health and wellbeing of people living with dementia. It is important that experiential elements are reported, as VR research often relies on quantitative methods of data collection that may lack an experiential focus. This paper added qualitative and experiential findings which as outlined by Braun and Clarke [61], can provide a more nuanced understanding of the causal mechanisms underpinning experiences and perceptions; in this context, underpinning VR technology probes and more broadly, VR use in general.

The findings suggest that VR may have a positive impact on the social health of people living with dementia. When using VR, a triadic relationship became apparent between people living with dementia, their caregivers, and the researcher. This was evidenced through increased verbal and non-verbal responses, as observed and reported during VR use. This relationship was also observed by Goodall et al. [67], who trialled SENSE-GARDEN, an immersive multisensory reminiscence experience for people living with dementia. Despite their memory difficulties, people living with dementia communicated their VR experience with the caregiver and the researcher during and after use. The structure of the content, like that discussed by Goodall et al. [67] facilitated a “flow of conversation” (p. 15). This is further reiterated by Muñoz et al. [68], who found that interactive elements in their iPad application served as a ticket to talk between people living with dementia, facilitators, and their family members [68]. This strengthens calls in the HCI community for increased research exploring how digital technology such as VR may enhance or maintain the social connectedness of people living with dementia [9,14]. It is noteworthy that this engagement and interaction may be reliant on adequate facilitation during VR use as previously acknowledged by Muñoz et al. [68] and Flynn et al. [14].

Consistent with previous research in this area [55], the HMD was well tolerated by all people living with dementia and caregivers. Applying the findings of Karaosmanoglu et al. [55] to this study, the adaptation of the controller to use one main button served as a welcome design choice for people living with dementia. Although additional facilitation was required during use, the use of just one button dissipated some of the anxiety surrounding the controllers for people living with dementia. The use of the comfort head strap and silicone eye piece increased comfort during use and are consistent with the procedures adopted in other VR exergame studies with older adults [69,70].

This study complements Koh et al. [9] acknowledgement that technology design is often not aligned with the needs of people living with dementia and their caregivers and the need to consult with practitioners such as OTs who have “knowledge of the holistic and occupation-based model to enhance the comprehensiveness of considerations for technology design in relation to people living with dementia” (p. 5). The inclusion of passive and active VEs within VR FOUNDations and its graded implementation is attributed to its health and social care-informed design. OT involvement has been advocated to inform technology design which interacts in a manner that is suited to the needs of everyone [9,71,72]. OTs are in a unique position to assess and identify the ecology of each person living with dementia by holistically assessing how they are interacting and responding to stimuli in the environment such as music, storytelling, etc. or gaining an understanding of the person’s life history [72,73,74]. This enabled people living with dementia to experience VR at a level that was suited to their needs and fostered the autonomous use of VR FOUNDations.

This study not only included the viewpoint of people living with dementia but also emphasised their caregivers’ contribution and acknowledges the wider sociotechnical system influencing VR implementation for people living with dementia. Although the focus of VR FOUNDations was to ensure accessibility for people living with dementia, their caregivers also brought useful insights into how people living with dementia may experience VR from a caregiver lens, providing practical suggestions for improving accessibility and reducing reliance on the caregiver when using VR FOUNDations. This research mirrors calls from Karaosmanoglu et al. [55] and Waycott et al. [75] to place formal and informal caregivers at the centre of the design process, seeking their contribution on how caregivers may be supported to set-up and facilitate future VR use.

The research team advocate for adequate training and resources for the caregivers as they are often required to assist with VR set-up and use. VR developers and researchers need to translate materials and set them up in the most convenient manner to ensure adoption and acceptance by the caregiver. It is vital that as researchers in this area we ’support the supporters’ [14]. This is consistent with Waycott et al. [75], who argue that an ethics of care framework should be used to emphasise the importance of care practices and staff to make VR safe and enriching for older adults in care settings. This is particularly important for VR implementation that is not completed by members of the research team [75].

The findings from this study reveal that VR FOUNDations can (1) increase people living with dementia and their caregivers’ understanding of VR and (2) provoke inspiration for future design decisions. These are the two main intents of technology probes. VR FOUNDations provided a means of exposing people living with dementia and their caregivers to VR, and in particular its interactive features. Interestingly, people living with dementia and their caregivers had various experiences using VR previously, for most only to passively interact through 360-degree video which did not enable full interactivity. This novel and sporadic exposure to VR is consistent with previous review findings [14]. Thus, people living with dementia and caregivers had presumptions about the interactive capabilities of VR. By experiencing the spectrum of interaction (both passive and active VEs), people living with dementia and caregivers reported a more nuanced understanding of VR and a grounding with which to inform the design of a bespoke VR application to support health and wellbeing (social health in the case of this larger project). It may also ensure that people living with dementia are appropriately introduced to VR before using more bespoke VR applications which focus on health and wellbeing outcomes.

VR FOUNDations served as a vehicle for knowledge mobilisation, whereby people living with dementia and their caregivers were supported in demonstrating their capabilities and were given a platform to share their lived experience through the medium of VR FOUNDations. Indeed, VR FOUNDations enabled this knowledge exchange in a manner that traditional, one-sided methods of data collection alone may not permit. When partaking in research activities, the phenomenon of interest must be clear for people living with dementia and their caregivers; VR FOUNDations provided a tangible artefact that could be experienced and discussed, which textual narratives alone could not provide. By using this approach, people living with dementia, caregivers and the research team established a mutual understanding as each were discussing the artefact and had exposure to its capabilities, albeit at a basic level. In the case of the larger PAR project, VR FOUNDations was considered an important prerequisite to the participatory design of a VR application to support the social health of people living with dementia.

Through exposure to the basic interactive capabilities of VR, people living with dementia and their caregivers openly discussed future design ideas suited to their dynamic and individual needs. People living with dementia and their caregivers were encouraged to view passive and interactive VEs and expressed interest in trying VR again in the future. It also instilled confidence in people living with dementia and their caregivers to use VR again independently. In participatory design, people living with dementia and caregivers must be empowered to meaningfully complete participatory methods of data collection to drive the research and avoid being tokenistic. People living with dementia and their caregivers were forthcoming in discussing future design preferences and experiences despite not being directly questioned about this. A recurring characteristic across suggested activities was escapism and exploration of the past, present, or future. Examples highlighted the need for meaningful VR activities that differ from familiar day-to-day routines. This suggests that people living with dementia and their caregivers anticipated future use and had an adequate understanding of the technology probe to expand on this and suggest other VR activities. Interestingly, people living with dementia contributed twice as many design recommendations than their caregivers, suggesting that VR FOUNDations empowered people living with dementia to meaningfully inform future VR design.

## 5. Implications for Future Research

The findings of this study lead to recommendations for the design of future VR technology probes targeted at people living with dementia and their caregivers, and provide an example of how VR technology probes such as VR FOUNDations may be applied in dementia care contexts. The findings from this study will inform a larger PAR project which aims to design a bespoke VR application to enhance the social connectedness of older adults living with dementia. Such a technology probe may be used as a prerequisite to participatory VR design, which addresses wider health and wellbeing and empowers people living with dementia to meaningfully participate in design methods by drawing on their experience of using VR FOUNDations. The promising positive experiences of using VR FOUNDations suggest that future bespoke applications, which specifically address health and wellbeing outcomes can build on this area. This study demonstrates the importance of this prerequisite stage.

People living with dementia and their caregivers valued the emphasis placed on a sensitive and graded introduction to VR which contributed to their overall positive VR experience. The findings reveal some key recommendations for the use of VR technology probes for people living with dementia that can inform future design and implementation; these are presented in Table 5. These recommendations may be used by future researchers to guide their design and implementation of VR technology probes for use with people living with dementia and their caregivers from the outset in health and wellbeing domains. Outside of informing technology probe research, the findings also influence broader VR design research by providing valuable insights into how a VR technology probe may be used in dementia care research.

## 6. Limitations

This study was completed with a small sample of nine people living with dementia and nine caregivers which may impact the generalisability and transferability of the findings. Nevertheless, the findings revealed rich qualitative data. The practicalities of recruitment during COVID-19 also posed issues for recruitment and achieving a diverse sample regarding sex, age, and VR technology experience. Future work could explore the use of technology probes with a larger sample size. The variance in the use of VR in the home may also be considered a limitation as some people living with dementia and their caregivers did not have a swivel chair in their home, and this impacted their range of motion within the VEs.

## 7. Conclusions

This work evaluated the use of a VR technology probe for older people living with dementia and their caregivers. The results indicate that a VR technology probe may be a useful prerequisite to the participatory design of VR applications for health and wellbeing. A VR technology probe can act as a familiarisation tool, assisting people living with dementia and their caregivers to understand basic interactions in VR, dissipating its novelty effect. After exposure to the VR technology probe, people living with dementia and their caregivers reported feeling empowered to make future design decisions and orientated to VR use, sparking interest and ideas for the future design of a social health application. Although the purpose of the VR technology probe was not to evaluate its usability or design aspects, the experiential aspects of using the probe for people living with dementia and their caregivers provides useful guidance for future VR design and shows promise in health and social domains. It is anticipated that this paper will prompt researchers to consider how best to orientate people living with dementia to VR from the conceptualisation of VR studies for health and wellbeing.

## Figures and Tables

**Figure 1 ijerph-19-16343-f001:**
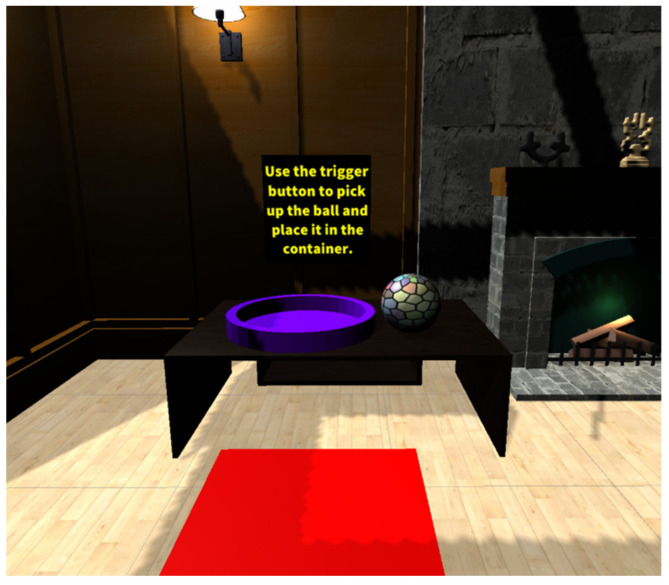
Image of a station with a ball and bucket including verbal prompt.

**Figure 2 ijerph-19-16343-f002:**
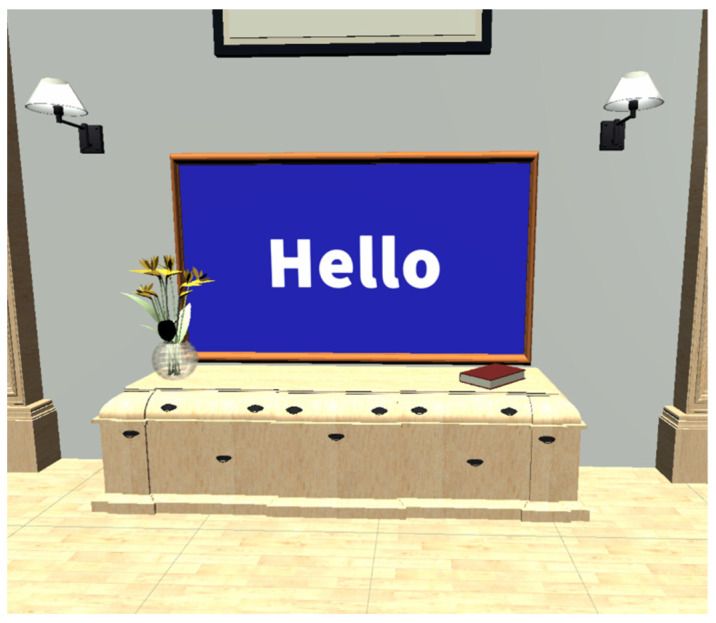
Image of a television to demonstrate video and sound in VR.

**Figure 3 ijerph-19-16343-f003:**
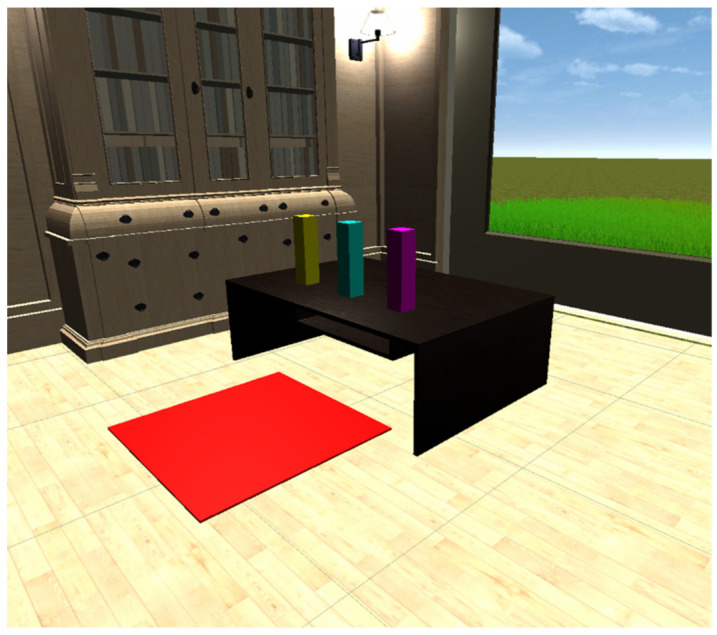
Image of a station in VR which enabled users to pick up, grasp and move blocks.

**Figure 4 ijerph-19-16343-f004:**
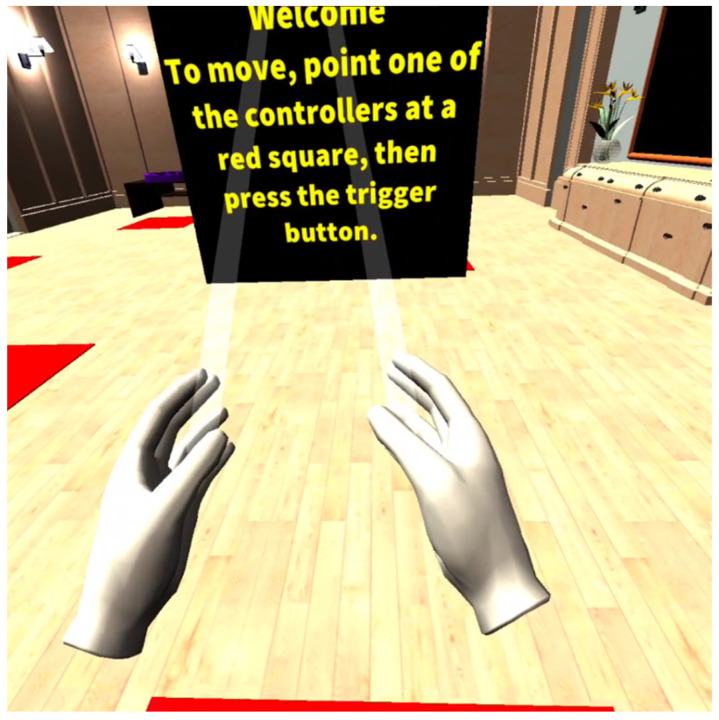
Virtual hands as represented in the VE.

**Figure 5 ijerph-19-16343-f005:**
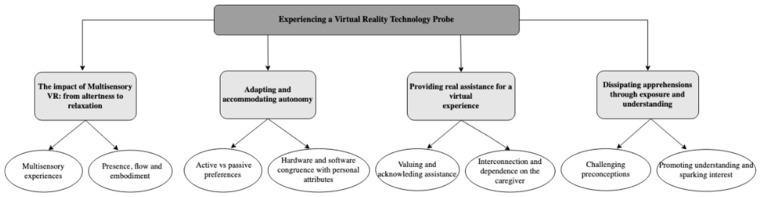
Thematic map of findings illustrating the overarching themes and corresponding subthemes.

**Table 1 ijerph-19-16343-t001:** VR FOUNDations Interactions.

Interactions	Description	Implementation
Selecting an item	Three coloured blocks are presented which can be selected, picked up and placed back down on the table. The virtual hands have a ray extending out which enables the user to see where to point.	Three cubes of different colours were added to the top of a table. The colours of the cubes are bright, contrasting with the dark colour of the tabletop. The outline of the cubes is highlighted when the user points at them and haptic feedback is provided when the user hovers over the cube.
Grabbing and placing an object	The user is presented with a ball and a container. The user can select the ball, pick it up and place it inside the container.	A ball and container were added to a table, both having bright colours distinct from the colour of the table. The ball is highlighted when the user points at it. The task is completed once the ball is placed within the container.
Moving to another location	The user can navigate the VR space by pointing at teleport squares on the floor. Once the teleport square is highlighted, they press the trigger button and are then moved to that new location indicated by the teleport square.	Teleport squares have a distinct colour compared to the floor. The colour of the teleport square changes when the user points at it. A pointer also appears at the position to which the user is pointing. After the user highlights the teleport square and then presses the trigger button, the screen fades out, the user is re-located, and the screen fades in afterwards.
Look at and hear	The user is presented with a screen in front of them and is required to watch the video until it is finished.	A video player contains a video the user can watch. The screen colour contrasts with the wall colour. The video only plays while the user is looking at the screen. The task is complete when the video reaches the end.

**Table 2 ijerph-19-16343-t002:** People living with dementia (PwD) Demographic Characteristics.

Cases	Gender	Age Range	Current Primary Caregiver	Length of Time Experiencing Memory Problems
PwD1	Male	59–69	Spouse/Partner	1–3 years
PwD2	Male	59–69	Spouse/Partner	4–6 years
PwD3	Male	70–79	Daughter	4–6 years
PwD4	Male	70–79	Daughter	1–3 years
PwD5	Female	80+	Daughter	7+ years
PwD6	Male	59–69	Spouse/Partner	1–3 years
PwD7	Male	70–79	Spouse/Partner	1–3 years
PwD8	Male	59–69	Spouse/Partner	4–6 years
PwD9	Female	70–79	Daughter	1–3 years

**Table 3 ijerph-19-16343-t003:** Caregiver (CG) Demographic Characteristics.

Case	Gender	Age Range	Relationship of PwD	Length of Time Supporting PwD
CG1	Female	50–59	Spouse/Partner	0–4 years
CG2	Female	50–59	Spouse/Partner	5–9 years
CG3	Female	40–49	Father	0–4 years
CG4	Female	40–49	Father	0–4 years
CG5	Female	50–59	Mother	5–9 years
CG6	Female	50–59	Spouse/Partner	0–4 years
CG7	Female	60–69	Spouse/Partner	0–4 years
CG8	Female	60–69	Spouse/Partner	5–9 years
CG9	Female	30–39	Mother	0–4 years

**Table 4 ijerph-19-16343-t004:** People living with dementia (PwD) and caregiver (CG) experience using technology and VR.

Experience Using Technology		Experience of VR	
A lot of experience For example: using a tablet, games console, laptop	5 PwD (PwD 1, 4, 6, 7, 9) 9 CG (CG 1–9)	Seen VR used but not used personally	4 PwD (PwD1,3, 7,9) 1 CG (CG 9)
Some experience For example: using a mobile telephone	3 PwD (PwD 3, 5, 8)	I have tried VR myself	3 PwD (PwD 2, 4, 6) 4 CG (CG 2–4, 6)
No experience	1 PwD (PwD 2)	No experience	2 PwD (PwD 5, 8) 4 CG (CG 1, 5, 7, 8)

**Table 5 ijerph-19-16343-t005:** Design and implementation recommendations for VR technology probes.

Theme	Derived Recommendation
The impact of Multisensory VR: from alertness to relaxation	Clear graphics with relatable content. Incorporate meaningful content to spark interest and curiosity. Provide a pleasant and welcoming aesthetic. Use bright and contrasting colours. Avoid sensory overload and overstimulation when designing VEs. Allow for natural and intuitive interaction and manipulation in the VE (through real time responsive virtual hands, rays, one-button selection). Incorporate relatable sounds relevant to the visual content. Accommodate for sensory difficulties (glasses, hearing aids etc) during the design. Provide an opportunity for engagement and relaxation through a library of activities.
Adapting and accommodating autonomy	Design for free exploration (no time limit or aim of task mastery). Provide opportunities for passive and interactive use: ○ One main controller (trigger) button. ○ Clear pointer and ray in the VE to illustrate where one is pointing/aiming. ○ Seamless movement in the VE with no latency. ○ Ability to move freely but also have some guiding points within the VE for those who need it. Design for dynamic functional abilities of people living with dementia. Accommodate seated and standing VR. Emphasis on free exploration and time set aside for people living with dementia and their caregiver to ask questions throughout the experience. Provide visual and verbal instruction within the VE.
Providing real assistance for a virtual experience	Ensure user eligibility before use. Clear explanation of the purpose of a VR technology probe. Clear communication and instructions (verbal and written). Active monitoring of verbal and non-verbal responses to VR. Provide a set-up manual and troubleshooting guide to support the use of VR for people living with dementia and caregivers. Real assistance to accompany virtual assistance tools. Support facilitators with education, set-up and monitoring advice throughout the process.
Dissipating apprehensions through exposure and understanding	Acknowledge the relative novelty of VR from the outset. Discuss the presumptions of VR with people living with dementia and their caregivers before use. Have the caregiver present when people living with dementia are using VR for reassurance and support. Provide opportunities to use technology probes from the outset of VR design projects. Provide technology probes which accommodate both passive and active experiences. Use technology probes to provide a grounding and understanding of VR to inform the design of bespoke VR applications. Involve PPI groups in designing probes, piloting early iterations of design and trialling the implementation procedures.

## Data Availability

Not applicable.

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
