# Peer review of "Introducing and Familiarising Older Adults Living with Dementia and Their Caregivers to Virtual Reality"

_ijerph, 2022, doi:10.3390/ijerph192316343_

Round 1

Reviewer 1 Report

This manuscript discusses the implementation and evaluation of a Virtual Reality technology probe involving people living with dementia and their caregivers. The paper is well-written and interesting. The following recommendations can improve the paper.

INTRODUCTION

- More explanation of what a technology probe is needed. How is a technology probe different from an early usability or a feasibility study? Moreover, what does it mean for a technology to be designed as a technology probe? Why existing VR headsets cannot be used as technology probes for older adults? What are they missing?

MATERIALS AND METHODS

- The design and development of VR Foundations should be moved to the materials and methods section

- Occupational Therapist on page 3 should be abbreviated as OT - not sure why it is AF

- A definition of passive VE should be added.

- It is not clear not whether the passive environment alone was used as a separate probe, and whether the interactive elements were post passive environment probe study or not.

- Table 1 - why and how these interactions were selected?

- Table 4 - also add the number of people with dementia and SP along with their identifiers.

Procedure

- What was order in which participants investigated the technology probe. Was it frustrating for the participants to wait for their turn. Moreover, they could have been influenced by each other's comments.

- What kinds of questions participants were allowed to ask? what kind of assistance were the facilitators allowed to provide during the test?

Data Analysis

- What is the PAR group? I believe it was never mentioned before. How are they relevant to the study?

- An explanation of the PAR cycles is needed.

RESULTS

- What is the adaptive technology probe?

DISCUSSION

- Not really sure what the authors are referring to when they mention user-centered design process in the discussion page 13 second paragraph. No user-centered design process that was undertaken prior to the study has been described.

Design Implications

- Table 5 - sources of the design principles are not provided.

Reviewer 2 Report

Authors have described a study that implements and evaluates a VR technology probe with nine people with dementia and nine informal caregivers. In general, this manuscript is well-written but does not fit well into the special issue since it does not explain how this system would benefit the health and well-being of people with dementia. I feel that this work deserves to be published, but at a different venue or even in this one if major revisions are made. Currently, the study has some weaknesses that greatly reduce its scientific value. Below, I offer several thoughts for the authors' consideration.

1. The manuscript should be proofread to reduce errors. For example, on Page 2 and line 52, the word design is spelt with a dash “de-sign”.

2. The abbreviation for people with dementia in the general bibliography is either PWD or PwD. It would be better if the authors used these abbreviations instead of PLWD to be consistent with the general literature.

3. It would be helpful for me to understand what a support person is responsible for. Does it fall under the informal caregiver category?

4. Considering this is not an HCI journal/conference, I suggest the authors downsize the HCI focus on the introduction and related work. In its current form, the manuscript would be a better fit for the HCI International Conference or the International Journal of Human-Computer Studies.

5. It is a bit difficult to follow the structure of the manuscript. Sections 1.4 and 1.5 should be moved to the methodology section, as they are no longer literature reviews or related work, but rather design and implementation.

6. Could you please describe the level of dementia your participants have?

7.  Adding subthemes would benefit the reader. A table or schematic figure will improve the clarity of the manuscript when the themes and subthemes of the analysis are presented.

8. The importance of the study is unclear to me. Although it claims to be an HCI paper presenting a co-design process, this is not fully explained. In what ways did people’s with dementia and caregivers contribute to the design? What changes have been made to the system based on the inputs of the users? In general, it seems that this is a simple study examining the use of VR with people with dementia and their caregivers.

9. This special issue aims to promote the health and well-being of people with dementia. The manuscript does not clearly describe this. You may wish to spend some time transforming the manuscript in order to make it more suitable for the call for papers for the special issue.
